# Endodontic Clinical Diagnostic Skills amongst Undergraduate Dental Students: Cross-Sectional Study

**DOI:** 10.3390/healthcare10091655

**Published:** 2022-08-30

**Authors:** Mohammed A Alobaoid, Omir Aldowah, Mohmed Isaqali Karobari

**Affiliations:** 1Department of Restorative Dental Sciences and Department of Dental Education, King Khalid University College of Dentistry, 3263, Abha 61471, Saudi Arabia; 2Prosthetic Dental Science Department, Faculty of Dentistry, Najran University, Najran 11001, Saudi Arabia; 3Department of Restorative Dentistry & Endodontics, Faculty of Dentistry, University of Puthisastra, Phnom Penh 12211, Cambodia; 4Department of Conservative Dentistry & Endodontics, Saveetha Dental College & Hospitals, Saveetha Institute of Medical and Technical Sciences University, Chennai 600077, Tamil Nadu, India

**Keywords:** clinical endodontics, cross-sectional study, diagnosis, education, undergraduate

## Abstract

The purpose of the study was to evaluate the clinical endodontic diagnostic skills amongst undergraduate dental students at pre-clinical and clinical levels at Cardiff University School of Dentistry. An online questionnaire containing eight questions about endodontic diagnosis and hypothetical clinical scenarios was sent to all year 3rd, 4th, and 5th-year undergraduate dental students who were divided into G1, G2, and G3 groups. The data were analysed descriptively and reported in percentages. Around 121 students out of 226 responded to the questionnaire with a response rate of 53.5%. The overall correct response from G1 (3rd year) was 31.6% to 65.8%, G2 (4th year) was 73% to 93%, and G3 (5th year) was 73.2% to 92.7%. The study concludes that the 4th and 5th-year undergraduate dental students’ responses to the hypothetical clinical scenarios were higher than the 3rd-year students. However, regarding questions about the endodontic diagnosis, the percentages of correct answers were similar among all the 3rd, 4th, and 5th-year students. Therefore, further studies assessing endodontic diagnostic skills amongst the same cohort of students during their progression in the undergraduate course are recommended.

## 1. Introduction

Diagnosing pulpal and periapical disease is a complex process that relies on several factors, including the chief complaint, medical, dental, and social history, clinical examination, diagnostic tests, and radiographic examination. A clinician must follow a systematic diagnostic evaluation process to achieve an accurate diagnosis. Endodontic clinical diagnosis is challenging, specifically when the results of sensibility tests are inconsistent with the patient’s complaints [1]. Thus, correct diagnosis appears based on the clinician’s adequate and updated knowledge and efficient clinical reasoning since errors encountered during this process may lead to misdiagnosis [2]. Therefore, when diagnosing pulpal and periapical diseases under clinical conditions, all available information and analysis should be employed to reduce the possibility of false-positive or false-negative errors and arrive at an evidence-based best clinical diagnosis [3]. In addition, dental students should be extensively trained in the knowledge of diagnostic procedures, examination processes, and tests and aware of their limitations. The dental students must develop their clinical skills to differentiate and accurately diagnose an endodontic disease associated with symptomatic and asymptomatic teeth. The most important of these assets are knowledge, interest, intuition, curiosity, and patience [4,5]. During undergraduate training, dental students are required to confirm their diagnosis and treatment plan with their supervisors before they execute the treatment. However, after graduation, they should be competent to make the most accurate endodontic diagnosis and treatment by themselves. There is only scarce and limited literature available for assessing undergraduate dental students’ diagnostic skills and deciding whether they improve during their training or not. This study aims to evaluate undergraduate dental students’ clinical endodontic diagnostic skills when presented with endodontic clinical scenarios at different levels of their Dental Surgery course [6]. The study also shows if clinical endodontic diagnosis skills improve as an undergraduate student goes through varying levels of training.

## 2. Materials and Methods

The current research was a descriptive, cross-sectional, questionnaire-based online study which included undergraduate dental students in their 3rd, 4th, and 5th years of BDS at the School of Dentistry, Cardiff. The Dental School Research Ethics Committee (DSREC) approved the study with reference number 1829a. The students were divided into G1, G2, and G3 based on their year of BDS. G1 included 70 third-year undergraduate dental students, G2 had 70 fourth-year undergraduate dental students, and G3 included 72 fifth-year undergraduate dental students. The questionnaire was designed and consisted of eight multiple-choice questions. The first three questions recorded general information about endodontic diagnosis procedures. The remaining five questions included hypothetical clinical case scenarios, where the participants were asked to state their differential diagnosis, identify the need for further special investigations if needed, and provide a definitive diagnosis. All the questions were posed in the multiple-choice format; however, for some questions, the participants were provided space to write another diagnosis, condition, or procedure if it was not listed among the options.

This online questionnaire was emailed individually by the principal researcher via the Cardiff Dental School and Hospital Undergraduate Students Centre to all three groups included in the study. Online Surveys (formerly Bristol Online Surveys (BOS); https://www.onlinesurveys.ac.uk/, accessed on 1 April 2019) were selected for data collection in this study as they are secure and fulfil all the requirements for online surveys. All participants were given information about the research study in the invitation email, and their participation was voluntary. Consent was obtained from all participants in the first part of the online survey.

The online survey was opened to all participants on 15 November 2018 and closed on 30 January 2019. Reminders were sent approximately every 2 weeks between the opening and closing dates to encourage participation and increase the response rate. In addition, avoiding obtaining or disclosing the student’s personal data to the researcher without their consent is against the General Data Protection Regulation (GDPR 2018).

The data was collected and analysed descriptively. The percentages were used to describe frequencies and represented in graphs and tables for a more straightforward interpretation. The comparison was made between the different levels of undergraduate dental students from years three, four, and five. The data were analysed descriptively, as the dependent variables (correct answers) in most questions are multiple, with three or four correct answers.

## 3. Results

In this study, 121 students at the School of Dentistry, Cardiff University, replied to the questionnaire with a response rate of 53.5%. Overall, 43% of the participants were male. Around 38 (31%) students from the 3rd year, 42 (35%) students from the 4th year, and 41 (34%) students from the 5th year participated in the study. A total of three questions on general endodontic diagnosis were presented to the students. It was found that more than 90% of the students responded that chief complaint, dental history, pulp sensitivity test, and periapical radiographs are essential steps in establishing an endodontic diagnosis. Around 81% of the students considered transillumination to be a necessary tool for attempting to locate cracks in teeth, while only 35.5% of students considered pulp sensitivity tests helpful in detecting cracks in teeth. Around 100% of the students considered ethyl chloride the best option to carry out a cold sensitivity test.

According to case scenario 1 as shown in Figure 1, the correct response of the differential diagnosis was reversible pulpitis and dentine hypersensitivity. Around 89% of the respondents in G1, 76% in G2, and 83% in G3 marked reversible pulpitis as the correct answer, whereas dentine hypersensitivity was selected by 32% of the respondents in G1, 81% in G2 and 53% in G3. In the second part of the question, it was found that 75% and 78% of students in the G3 groups selected thermal and electric pulp testing, respectively. This is much higher than the G1 (47%) and G2 (57%) groups for thermal tests and the G1 (71%) and G2 (71%) groups for electric pulp tests.

The correct answers to the possible causes of the tooth discolouration were either pulp necrosis or previous root canal treatment of the tooth, as shown in Figure 2. Pulp necrosis was selected as the highest option in all groups, with 65.8%, 93%, and 73.2% for G1, G2, and G3, respectively. In contrast, previous root canal treatment of the tooth was selected by only 13.2% of G1, compared to 50% and 68.3% for both G2 and G3, respectively. The second part of the question asked for a systematic approach to making a definitive diagnosis. Again, the majority of the participants in all three groups selected the same option, presenting complaint history, dental history, medical history, special tests, clinical examination, and differential diagnosis with a response rate of 86.8% in G1, 93% in G2, and 95% in G3 groups.

According to case scenario 3, as shown in Figure 3, the correct answer was Chronic Apical Abscess, which was selected by 73.8% of students in G2, 75.6% in G3, and only 42% in G1 group. Regarding the participants’ selection of the correct option to detect the origin of the sinus tract, 95.2% of the G2 and 92.7% of the G3 groups selected the most feasible answer, which was pulp sensitivity testing plus inserting a gutta-percha cone into the draining buccal sinus tract and taking another periapical radiograph, compared to only 55.3% of the G1 group.

According to case scenario 4, as shown in Figure 4, 85.7% and 83% of students from G2 and G3 selected the correct answer: Symptomatic Irreversible Pulpitis with Normal Apical Tissue, compared to 42% of students in the G1 group. Regarding the treatment plan for this clinical scenario, 48.8% of students in group G3 chose the most realistic answer: primary root canal treatment without replacing the extra-coronal restoration. This was followed by 35.7% of students in group G2 and 15.8% in group G1.

According to case scenario 5, as shown in Figure 5, all the tools can be used to reach an initial diagnosis except for Laser Doppler Flowmetry. A high percentage of students from the G2 (90%) and G3 (91%) groups chose percussion and palpation compared to EPT and thermal testing. Approximately 87.8% of students from G3, 78.6% from G2, and 55.3% of the G1 group selected the best answer: previously treated tooth with Symptomatic Apical Periodontitis.

## 4. Discussion

The undergraduate course at Cardiff University’s dental school is a five-year program. To complete the dental curriculum, students must be competent in academic concepts, clinical knowledge, and interpersonal skills [8]. Endodontic treatment may be a stressful procedure for undergraduate students [9]. Meanwhile, it is additionally the procedure that is performed most commonly in clinical practice. It typically calls for proficient technical abilities, experience, and knowledge of pulp anatomy and its variants [10]. As a result, undergraduate students need to have a high level of self-confidence in both their theoretical knowledge and their clinical abilities in endodontics. Undergraduate students’ self-evaluation would be a valuable tool for objectively analysing the dental curriculum and determining which courses are most beneficial.

Therefore, this study aimed to determine how well undergraduate dental students could diagnose hypothetical clinical situations. The students were split up by their academic years to see if there were any differences in their abilities.

In this study, the total response rate was 53.54%. This agrees with an investigation by Aboalshamat et al. [11], where the response rate of undergraduate dental students was only 17%. However, the literature has not come up with an apparent minimum response rate, but some authors think that a response rate of 70–80% is enough to rule out non-response bias [6,12]. In contrast, a response rate of 50% is acceptable for questionnaires targeted toward undergraduate students [13]. Response rates can be different because of many things, such as the topic of the questionnaire, the difference between respondents and non-respondents, the size and number of the sample, the type of group being targeted, the way the questionnaire is made, and how it is given out [14]. The current study’s response rate was meagre (10%), even with the participants being reminded via email every two weeks since the survey was opened. In the published literature, the effect of reminders on increasing response rates is controversial, as some endorse its effectiveness [15] and others do not [16]. However, this study gave reminders by approaching the undergraduate dental students during their lectures, laboratory, or clinical sessions. In addition, representatives from each group were sent an email asking them to get their co-workers to participate in this study. Although this method was time-consuming, the effect was noticeable, as the response rate increased from 10% to 53.54% when the survey was closed.

To make a correct diagnosis, it is necessary to carefully analyse both the objective and subjective information obtained from the clinical pulp sensitivity tests and any other tests that may be necessary. Pulp diagnosis typically entails a thermal or electrical sensitivity test, which aims to excite the pulp by causing a fluid flow in the dentinal tubules [17]. The result of this study revealed that most undergraduate dental students in all three years were unaware of the various tools that could be used when carrying out cold sensitivity tests. A study with similar findings showed that 84% of undergraduate students either never utilised the pulp testing or used it only occasionally [18]. All students (100%) chose ethyl chloride, while only 8.3% and 0.8% considered Endo-Ice and ice sticks, respectively. This could be explained as ethyl chloride being the only available method for the students during their training. Although ethyl chloride provides a source for reducing the tooth temperature, it is no longer recommended because the temperature obtained is less effective than endo-ice, dichlorodifluoromethane (DDM), dry ice, and carbon dioxide snow [3,19]. Surprisingly, it was noticeable that the 3rd and 4th-year students were more aware of the importance of dental and medical history taken during diagnostic procedures compared with 5th-year students. The possible explanation could be attributed to the different abilities of each cohort of students in additional years. Past and current medical histories are essential to clinical diagnosis in dentistry [20]. A complete medical history is also necessary for disease management and clinical decision-making. For example, it would be preferable to avoid dental extractions or endodontic surgery in patients who have received bisphosphonate treatment with a high risk of osteonecrosis when such things are achievable [21].

In the hypothetical clinical case scenarios, it was noticed that undergraduate dental students (G1) chose the correct diagnosis less frequently compared to G2 and G3. For example, in the first clinical scenario of G1 participants, a small number of students (31.6%) considered dentine hypersensitivity in addition to reversible pulpitis as a differential diagnosis compared to 81.6% and 63.4% of G1 and G2 participants, respectively. The reason that year five’s correct answer percentage was lower than year four is unclear. The possible reasons could be attributed to the different abilities of students to recall information, lack of knowledge, or response bias, which means a lack of desire to answer the questions [15]. Many undergraduate dental students reported experiencing difficulty in endodontic diagnosis when they were in a clinical environment. Concerning, however, was the fact that a significant number of participants in the fifth year did not consider themselves knowledgeable in endodontic diagnosis. According to the feedback received from undergraduate students, they reported feeling incompetent as a direct result of having insufficient experience in clinical settings.

In the second clinical scenario, a small percentage of G1 participants considered a previous root canal treated tooth as a possible cause of tooth discolouration (13.2%), while the percentage of participants who selected this as an option increased in G2 and G3 (50% and 68.3%, respectively). Finally, in the third hypothetical clinical case scenario, the same observation was noticed, as only 16 out of 38 participants (42%) of G1 selected the correct diagnosis compared to 31 out of 42 (73.8%) and 41 (75.6%) of G2 and G3, respectively.

The same result can be observed when participants were asked about the clinical tests or examinations they would like to perform before making a definitive diagnosis for the first hypothetical clinical scenario. The participants were provided with options for thermal tests, EPT, palpation, periapical radiographs, and BPE. A few G1 participants would consider these options before making a definitive diagnosis. In contrast, G3 (year 5) had the highest percentage of participants who selected all correct possibilities before making a definitive diagnosis, followed by year 4 and then year 3. The possible explanation could be because of less exposure to clinical cases of G1 compared to the rest of the groups. The impact of early clinical exposure on undergraduate dental students’ learning experience has been evaluated [17]. The study used mixed questionnaires and interviews with undergraduate dental students and clinical supervisors. The authors concluded that both the students and their supervisors thought early clinical exposure was helpful because it gave students a context for their theoretical learning and helped them learn how to apply their knowledge in clinical practice, as well as improve their interpersonal skills and ability to work as a team [22].

The three groups’ percentages of correct answers were similar in the general questions about the endodontic diagnosis part. Surprisingly, it was noticeable that the G1 and G2 (44.7% and 57%, respectively) participants were more aware of the importance of dental and medical history taken during diagnostic procedures compared with G3 participants (36.6%). The possible explanation could be attributed to the different abilities of each cohort of students in additional years. Past and current medical histories are essential to clinical diagnosis in dentistry [20]. As a result, the patient’s medical history and the findings from the clinical examination were the crucial elements of information required to make an accurate diagnosis. Students were instructed to diagnose based on the results of visual intraoral examination and radiographs. In addition, students were given the assignment to collect all of the information they had gathered and develop an individualised treatment plan. The complete medical history taken is also essential for disease management and clinical decision-making. For example, it would be preferable to avoid dental extractions or endodontic surgery in patients who have received bisphosphonate treatment with a high risk of osteonecrosis when such things are achievable [21].

Generally, when presented with various clinical scenarios, in this study, it was comprehensively observed that the 3rd year undergraduate students chose the correct diagnosis less frequently compared to the 4th and 5th year students. The possible explanation for this could be the less exposure to clinical cases of 3rd-year students compared to the rest of the groups. A recent study evaluated the impact of early clinical exposure on undergraduate dental students’ learning experiences and concluded that the students and supervisors perceived early clinical exposure to help offer a context to students’ theoretical learning and develop their understanding of the application of knowledge in clinical practice, interpersonal skills, and team-working skills [22]. In this study, the percentage of the students who chose the correct diagnosis increased from the 4th to the 5th year. However, only a high percentage of 5th-year students could identify the correct treatment plan for a particular diagnosis. This can be attributed to the different abilities of students to recall information, lack of knowledge, or response bias, which means a lack of desire to answer the questions [15]. According to De Moor et al. [23], the curriculum should include elements of instructional teaching, pre-clinical operative technique classes, and clinical patient treatment; clinical endodontics should preferably be supervised by specialists or staff members with specialised knowledge and interest in endodontics; specific assessment procedures should be an integral part of the curriculum in endodontics, with both formative and summative assessment protocols; and philosophy of patient-centred care [23].

There are several limitations to this study. The research has been conducted at one school of dentistry, which means that the result is not representative of all dental schools. Additionally, this study’s response rate (53.54%) was lower than expected. Finally, the current study’s questionnaire was designed with multiple correct options for most of the questions. Therefore, the groups’ overall answers were difficult to compare. For this reason, a descriptive analysis method was used in this study.

## 5. Conclusions

From this study, it can be concluded that the endodontic diagnostic skills of undergraduate dental students improve as the year of training advances. The third-year undergraduate students presented with fewer skills when compared to the 4th and 5th-year students. Furthermore, it can be concluded that students should be kept up to date on current practical skills and knowledge in addition to the theory provided in textbooks. Obtaining the most accurate endodontic diagnosis is based on several factors, and clinical training is one of the most critical factors. It is essential to improve the quality of dental education by doing further research to evaluate and compare the diagnostic and treatment abilities of dental students attending various dental schools.

## Figures and Tables

**Figure 1 healthcare-10-01655-f001:**
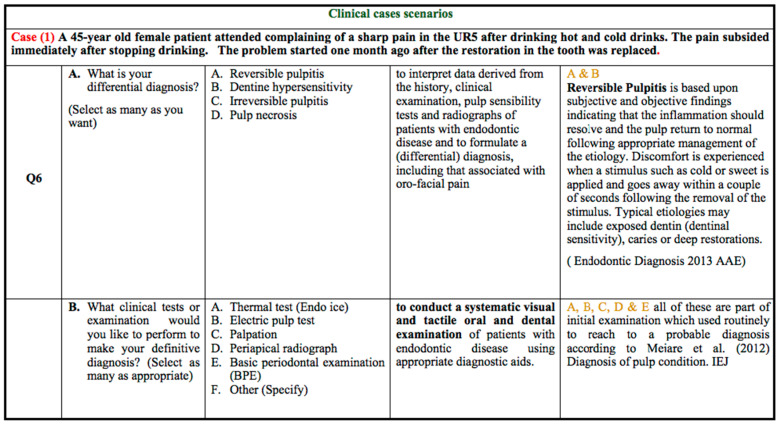
Clinical case scenario 1 [7].

**Figure 2 healthcare-10-01655-f002:**
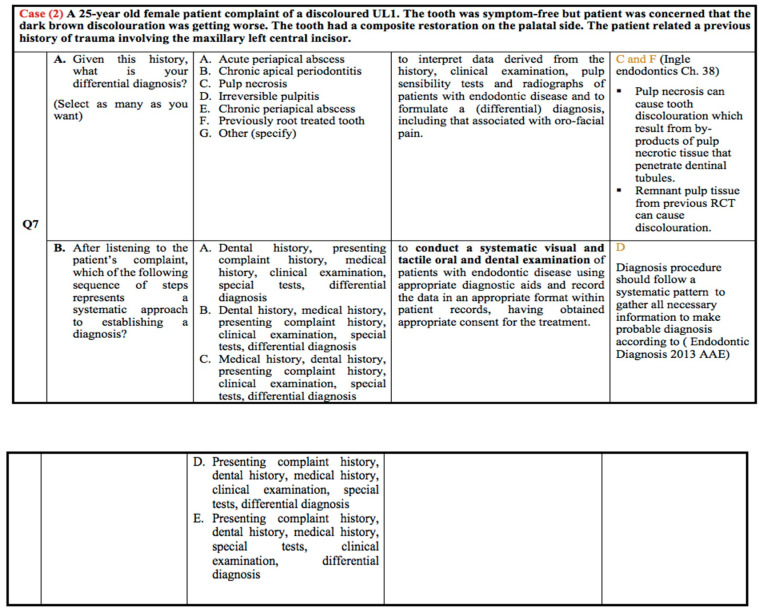
Clinical case scenario 2.

**Figure 3 healthcare-10-01655-f003:**
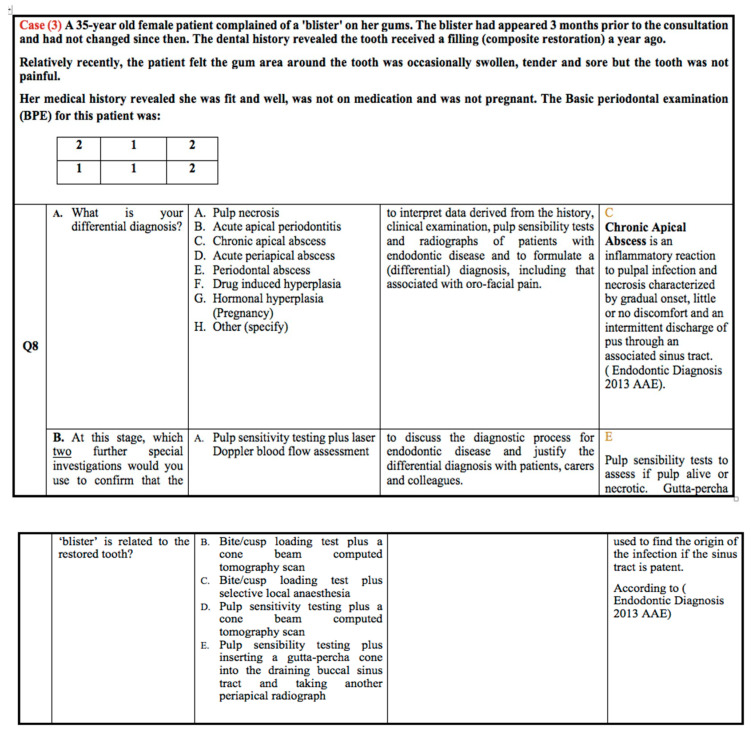
Clinical case scenario 3.

**Figure 4 healthcare-10-01655-f004:**
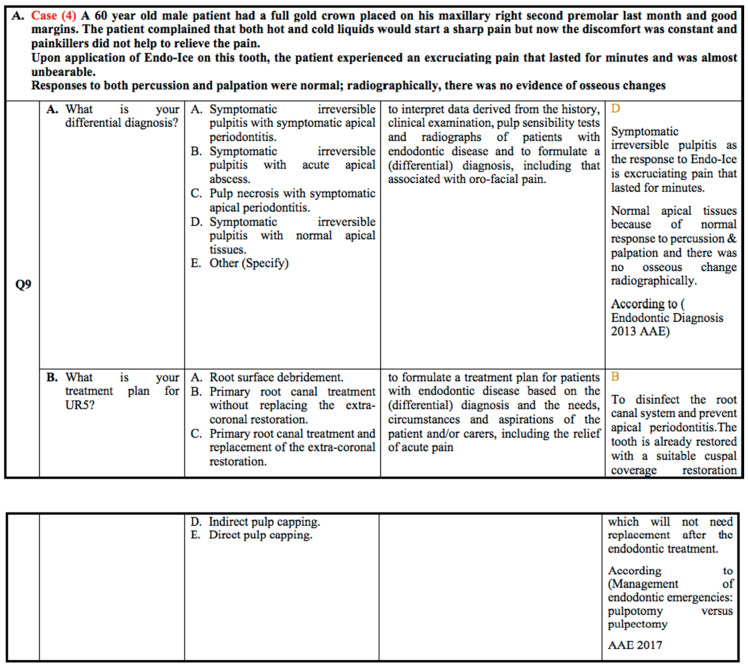
Clinical case scenario 4.

**Figure 5 healthcare-10-01655-f005:**
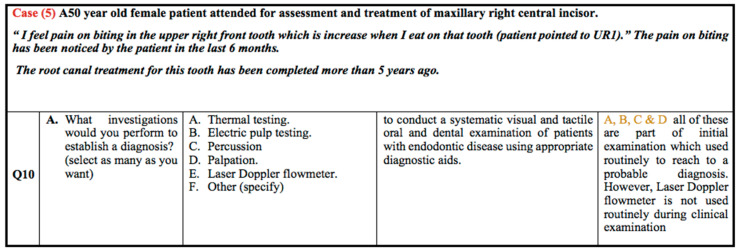
Clinical case scenario 5.

## Data Availability

Data will be made available from the corresponding author on reasonable request.

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
