# Peer review of "Endodontic Clinical Diagnostic Skills amongst Undergraduate Dental Students: Cross-Sectional Study"

_healthcare, 2022, doi:10.3390/healthcare10091655_

Round 1

Reviewer 1 Report

The idea of this manuscript is of interest, but the study is quite poor.

The authors don't report demographic data, but in scientific literature we know that female students have better performance in respect of the male ones.

However, no data are reported about the study period, and if the questionnaires were administered at the same time to all partecipants, during the course or after exam session.

The authors don'report statistical analysis because " It became more difficult to apply a statistical test especially when a participant selected one or two of the correct answers but not all of them.", but tests are available to validate such type of data.

The discussion is quite reudndant and confusing, and should be revised and better organised.

At last, English language and grammar must be revised. In addition, consider a lot of typos in the text (i.e., lines 62-65: compromised, and much more)

Author Response

Thank you for giving us the opportunity to submit a revised draft of our manuscript titled Endodontic clinical diagnostic skills amongst undergraduate dental students: Cross-sectional study to the Healthcare journal. We appreciate the time and effort dedicated by the editor and the reviewers to provide your valuable feedback on our manuscript. We are grateful to the reviewers for their insightful comments on our paper. We have been able to incorporate changes to reflect most of the suggestions provided by the reviewers. We have highlighted the changes within the manuscript in yellow colour.

Reviewer 2 Report

The comparison between different academic degrees entails bias, given that the clinical and theoretical experience of the different students varies, they must be more explicit in the scale or in the instrument (questionnaire) not all cases can be generalized because they can find false positives, not all pain means that it requires endodontics or that it is linked to this specialty, the general interest of the article is low, perhaps if they implemented a technique for diagnosis or based on their results they proposed something new, the article would be more attractive, the rest is average I would leave it up to the editor. In terms of bibliography, it can be improved, the most recent is from 2018, and the rest is one from 2016 and another from 2017, the others are very old, surely you can find some other more recent study.

Author Response

(The authors gave the same response as above.)

Round 2

Reviewer 1 Report

The authors replied satisfactory to my previous comments and suggestions